# Clomiphene Citrate Shows Effective and Sustained Antimicrobial Activity against *Mycobacterium abscessus*

**DOI:** 10.3390/ijms222011029

**Published:** 2021-10-13

**Authors:** Da-Gyum Lee, Yoo-Hyun Hwang, Eun-Jin Park, Jung-Hyun Kim, Sung-Weon Ryoo

**Affiliations:** 1Center for Clinical Research, Masan National Tuberculosis Hospital, Korea Centers for Disease Control and Prevention, Changwon 51755, Korea; sh050301@naver.com (D.-G.L.); hwangyoohyun@gmail.com (Y.-H.H.); dmswls090909@gmail.com (E.-J.P.); 2Infection Control Convergence Research Center, Chungnam National University College of Medicine, Daejeon 35015, Korea; 3Division of Intractable Diseases, Center for Biomedical Sciences, Korea National Institute of Health, Korea Centers for Disease Control and Prevention, Cheongju 28159, Korea; kjhcorea@korea.kr

**Keywords:** *Mycobacterium abscessus*, clomiphene citrate, drug repurposing, drug resistance, *non-tuberculosis mycobacteria*

## Abstract

*Mycobacterium abscessus* (*M. abscessus*) causes chronic pulmonary infections and is the most difficult non-tuberculous mycobacteria (NTM) to treat due to its resistance to current antimicrobial drugs, with a treatment success rate of 45.6%. Thus, novel treatment drugs are needed, of which we identified the drug clomiphene citrate (CC), known to treat infertility in women, to exhibit inhibitory activity against *M. abscessus*. To assess the potential of CC as a treatment for *M. abscessus* pulmonary diseases, we measured its efficacy in vitro and established the intracellular activity of CC against *M. abscessus* in human macrophages. CC significantly inhibited the growth of not only wild-type *M. abscessus* strains but also clinical isolate strains and clarithromycin (CLR)-resistant strains of *M. abscessus*. CC’s drug efficacy did not have cytotoxicity in the infected macrophages. Furthermore, CC worked in anaerobic non-replicating conditions as well as in the presence of biofilm. The results of this in vitro study on *M. abscessus* activity suggest the possibility of using CC to develop new drug hypotheses for the treatment of *M. abscessus* infections.

## 1. Introduction

*Non-tuberculosis mycobacteria* (NTM) are highly abundant in environmental niches such as soil and water sources, often leading to high human-pathogen contact [1]. In addition, several host factors, such as the ageing of the global population, lung diseases including cystic fibrosis (CF) and bronchiectasis, immunosuppressive, and broad-spectrum antibiotic therapy, contribute to the rise of NTM infections. NTM infections regularly surpass the global incidence of new tuberculosis infections in developed countries [1]. 

Among all NTM, *Mycobacterium avium* (*M. avium*) and *M. abscessus* represent the most frequent pathogens associated with pulmonary disease [2]. *M. abscessus* is a rapidly growing NTM with great clinical significance in CF patients, as those with *M. abscessus* infections have a more rapid decline in lung function. A *M. abscessus* infection can be an obstacle to subsequent lung transplantation, resulting in a poor clinical outcome or life-long persistent infection without symptoms [3,4,5].

*M. abscessus* is the most commonly isolated rapidly growing mycobacteria from lung infections, an alarming fact given the average rate of treatment success is only 45.6% [6,7]. For patients with chronic lung disease from NTM, none of the currently available treatments are curative nor have been effective in long-term sputum conversion [6]. Current treatment recommendations for *M. abscessus* pulmonary infections include combination therapy of two or more intravenous drugs (i.e., amikacin, tigecycline, imipenem, and cefoxitin) with one or two oral antimicrobials including classes of macrolides, linezolid, clofazimine, and occasionally a quinolone-derived drug.

Prolonged multi-antimicrobial therapy is often limited by drug-induced toxicity such as bone marrow suppression by linezolid, liver toxicity by tigecycline, development of hypersensitivity to β-lactams, etc. Even under strict regimens, treatment failure rates remain high with recurrent or chronic infections and grave clinical outcomes. There have been many studies recently on experimental antibiotics with potential activity against *M. abscessus* in various mechanisms and new therapeutic treatments. However, no work has been done on the actual efficacy of those treatments against *M. abscessus*.

There is a surge in research into experimental antibiotics with potential activity against *M. abscessus* in various mechanisms and new therapeutic treatments. However, there have not been sufficient studies for the efficacy against *M. abscessus*.

New NTM drug development is extremely slow and costly compared to its demand. Repurposing drugs previously verified for other diseases accelerate this development process with detailed safety data while avoiding many difficulties. Clomiphene citrate (CC), a drug known to treat infertility, has been shown to have antimicrobial activity against Gram-positive *Staphylococcus aureus*, *Bacillus subtilis*, and *Enterococcus faecium* [8,9]. CC targets the cytoplasmic enzyme undecaprenyl diphosphate synthase (UppS), which synthesizes carrier protein undecaprenyl phosphate (Und-P), the lipid responsible for transporting a significant substrate in wall-techoic acid (WTA) synthesis [8]. 

However, the mechanism of CC in *M. abscessus* is not yet well-researched. In this study, we report a detailed biological evaluation of CC against *M. abscessus*.

## 2. Results

### 2.1. CC Is Active against M. abscessus

The Clinical & Laboratory Standards Institute (CLSI) currently recommends the Mueller–Hinton (MH) microdilution broth-based method as the gold standard for determining the minimum inhibitory concentration (MIC) of antimicrobial agents for *M. abscessus* [10]. Therefore, we obtained the MIC of CC for *M. abscessus* in MH broth with resazurin rather than the 7H9 broth-based method. Additionally, we analyzed the dose-response curves of the CC for *M. abscessus* in MH and 7H9 broth. As shown in Figure 1A, the IC_50_ of CC for *M. abscessus* was 4.29 ug/mL in MH broth and 17.11 ug/mL in 7H9 broth, respectively. CC exerted similar potencies against reference strains of the *M. abscessus* irrespective of the type of the media MH or Middlebrook 7H9 broth.

We adapted reporter-based assays that are well suited for drug discovery applications owing to their simplicity and sensitivity compared to dye-based and absorbance-based assays [11]. We constructed luminescent reporter strains and then used them to determine the MICs of CLR and CC. Foremost, in vitro activities of CC and CLR were measured by reporter-based bioluminescence, and then the MIC data were obtained using the Resazurin Microtiter Assay (REMA) method. We used bioluminescent *M. abscessus* strains to validate our assays for drug susceptibility testing. CLR was used as a reference control. Dose-response curves of CC in *M. abscessus*-LuxG13 are shown in Figure 1C, and IC_50_ value is 5.358 ug/mL. Therefore, CC could be considered an effective drug candidate for *M. abscessus*. 

### 2.2. CC Is Active against Clinical Isolates of the M. abscessus and Claritromycin Resistant Mutant

Next, we determined whether CC holds this potent activity against a panel of clinical isolates, including rough (R)- and smooth (S)- colony morphotypes. CC was equally effective against all nine strains from the *M. abscessus* clinical isolates panel, with IC_50_ ranging from 4.44 to 6.90 ug/mL, similar to the MICs observed for the subspecies reference strains (Figure 2). The rough morphotype tends to be much more virulent than the smooth type and in the manner that it can resist host defense strategies [12,13]. These results demonstrated that CC was effective in vitro against the reference strain *M. abscessus* (ATCC 19997) and the clinical R- and S- colony morphotype strains.

We tested whether CC effectively inhibited the growth of drug-resistant strains that were laboratory-generated from this study at high concentrations (100 ug/mL) of CLR, which uses in anti-*M. abscessus* regimens. A laboratory-generated resistant mutant showed high drug resistance to CLR, as shown in Figure 3 and Appendix A. Furthermore, the CLR-resistant variant was susceptible to CC as the wild type with the same MIC range (3.35 ~ 4.439 ug/mL). It was the same regardless of whether it was a CLR-resistant or a susceptible strain. Thus, CC also works as an active inhibitory agent against CLR-resistant *M. abscessus*. 

### 2.3. CC Is Susceptible to Non-Replicating and Biofilm Growing M. abscessus

We ascertained the activity of CC against non-replicating phase cultures, this phase of which was induced by oxygen starvation. Before assessing the drugs′ effect, we confirmed the non-replicating condition by measuring the growth curves for *M. abscessus* under aerobic and anaerobic conditions (Appendix A). We compared the growth rate in each condition and recovered aerobic conditions in the same medium. The anaerobic state had a low growth rate, as we confirmed that the oxygen concentration had dropped to a lower level required for growth. Decolorization was observable after 15 h of adding the oxygen indicator dye, methylene-blue, to the anaerobic jar (Appendix A).

CLR showed increased activity against anaerobic cultured *M. abscessus* (IC_50_ 0.077.84 ug/mL) compared to the aerobic condition (IC_50_ 0.046 ug/mL), demonstrating a significant two-fold shift to higher IC_50_ values. (Figure 4). Interestingly, CC shows the lower IC_50_ for anaerobic cultured *M. abscessus* (IC_50_ = 3.157 ug/mL) than aerobic condition (IC_50_ = 4.592 ug/mL). Thus, CC gained some activity against anaerobic non-replicating *M. abscessus*. Furthermore, CC also attained some activity against anaerobic *M. abscessus*, closely related to the non-replicating environment.

Biofilm growing is an essential factor in antimicrobial resistance. Resistance to antibiotics, disinfectants, and germicides by biofilm-forming microorganisms may lead to treatment failure. Clinical experience has demonstrated that biofilms have to be physically eradicated to resolve the infection [14]. Muñoz-Egea et al., found differences between the MIC and MBEC in *M. abscessus*, specifically how MBEC of CLR increased 100,000 times compared to MIC [14]. However, we affirmed that when *M. abscessus* was exposed to CLR, the MBEC was 2000 times higher from 0.46 ug to >100 ug than the typical MIC of CLR.

As can be seen in Figure 4, CLR has excellent bactericidal activity (IC_50_ = 0.046 ug/mL) in growing culture but ultimately loses its activity against biofilm-*growing M. abscessus* (IC_50_ > 100 ug/mL). On the other side, CC still has an IC_50_ of 15.94 ug/mL against biofilm-growing bacteria. Thus, it seems that it has activity against biofilm-growing *M. abscessus*. Additionally, CC has enhanced action against oxygen-depleted non-replicating states of *M. abscessus* (IC_50_ = 3.157 ug/mL). These results suggest CC is an attractive substance for all kinds of anti-*M. abscessus* therapy.

### 2.4. CC Is Effective against Intracellular M. abscessus

It is common knowledge that pulmonary disease-causing NTM are successful facultative intracellular pathogens that survive and persist within the host macrophages [15]. This suggests that the research protocols searching for anti-NTM drugs should include methods to identify effectiveness intracellularly. We referenced macrophage assays engaged in studying the intracellular activity of the candidates against NTM [16]. We assessed the cell viability at different concentrations for determining whether CC influences cytotoxicity or not. This assay was assessed one day after the treatment using the Cellrix^®^ Viability assay kit and lactate dehydrogenase (LDH) assay kit. We presented the harmlessness of CC on the viability of human large intestine epithelial cells (HCT-8) and THP-1 cells in Appendix A. CC did not demonstrate cytotoxicity under particular concentrations that inhibited intracellular *M. abscessus*.

The use of the dual-readout assay allowed dose-response curves for both *M. abscessus* inhibition and THP-1 cells cytotoxicity determined in the same experiment [17]. Specifically, IC_50_ and CC_50_ (cell toxicity concentration that induces 50 % cell death) can be determined. Figure 5 shows that dual dose-response curves respond to CC, as the data obtained from the same screening wells over time. Cell toxicity was determined using a SYTO 60 probe. Tae sung et al. presented that clarithromycin dramatically decreased the number of intracellular *M. abscessus* present at 2 days after infection at concentration of 0.1 ug/mL [18]. Treatment with CLR successfully rescued the cells from the infection challenge and prevented bacterial replication (data not shown). CC inhibited *M. abscessus* (lux) growth in dual-readout assay with an IC_50_ of 2.294 ug/mL, without THP-1 cell toxicity even at the highest concentration (Figure 5A).

Consistently with the THP-1 infection model, CC showed a similar inhibitory effect on *M. abscessus*-GFP growth in THP-1, with an IC_50_ of 13.18 ug/mL (Figure 5B). Accordingly, we demonstrated the ability of CC to diminish intracellular *M. abscessus* infections.

## 3. Discussion

NTM infections are more common than tuberculosis in industrialized countries, and cases are likely to increase globally. Talking about diseases caused by NTM, especially involving *M. abscessus,* remains problematic [19,20]. It is very challenging to treat *M. abscessus* due to NTM’s resistance to most antibiotics, including macrolides, aminoglycosides, rifamycins, tetracyclines, and β-lactams. There is no adequate regimen of proven or predictable efficacy for *M. abscessus* lung disease. As an alternative, usual treatments consisted of macrolides (e.g., azithromycin or clarithromycin), aminoglycosides (e.g., amikacin), and beta-lactams (e.g., imipenem or cefoxitin). To get particular outcomes, patients have to be medicated for 18 to 24 months [21,22,23]. Unfortunately, notwithstanding combinational regimens, the success rate ranges only from 25% to 42% [24]. It has been shown that approximately 20% of *M. abscessus* isolates of pulmonary infections fail to respond to macrolides-included therapy [25,26]. Thus, there is an urgent medical need to discover and develop novel and more effective anti-*M. abscessus* drugs.

We could infer a candidate chemical, CC, that already has pharmacologic and toxicologic data. By such means of repurposing, existing drugs can be modified to fit a new use and shorten the drug development timeline [27]. Moreover, CC appears to have few reported adverse side effects [28,29]. CC is a nonsteroidal selective estrogen receptor and modulator that blocks estrogen receptors stimulating ovulation in anovulatory women [12,30]. Furthermore, CC had demonstrated efficacy against *Staphylococcus aureus* and *Bacillus subtilis* in vitro, with a MIC of 8 ug/mL, and incubation of *Bacillus subtilis* existing CC at the same MIC changed its morphology. Above all, it is very encouraging to report that CC is active against *Mycobacterium tuberculosis* H37Rv with MIC of 24.8 ug/mL, and synergy was observed when combining with macozinone in vitro [31,32]. In this study, we discovered the anti-mycobacterial activity of CC as a drug repurposing.

It is already known that the R morphotype tends to be much more virulent than the S morphotype [13]. R morphotype is involved in CF airways’ chronic colonization [12]. CC is effective in vitro against both the reference strain and the clinically isolated Rough (R) and Smooth (S) colony morphotype strains in our study. In this regard, CC has one additional advantage in clinical significance.

One of the things that makes the treatment of *M. abscessus* difficult is that it shows intrinsic resistance against CLR. It frequently happens in *M. abscessus* clinical isolates, especially in R-type clinical strains. In order to find out whether CC is effective against strains resistant to this important CLR, spontaneous CLR-resistant mutants were screened in this study. Furthermore, these CLR-resistant mutants show high drug resistance to CLR. According to our experimental results, as shown in Figure 2 and Figure 3, in vitro results suggest that CC is active to susceptible ones and induced CLR-resistant and clinical R morphotype *M. abscessus* strains. Our experimental results indicate that CC has a great advantage when used as a possible therapeutic drug for *M. abscessus* disease.

The ability of *M. abscessus* to produce biofilm represents a successful survival strategy for these ubiquitous microorganisms forming biofilm on the airways’ surface inside the human lung [33]. *M. abscessus* disease progression shares some aspects with *M. tuberculosis*. As the disease evolves, this pathogen also lives inside granulomas or pulmonary nodules characterized in anaerobic conditions [34,35]. This behavior is linked to their pathogenicity and their increased tolerance to antimicrobials [36,37]. Generally, the drug activity is established against aerobically growing active *M. abscessus*. Thus, treatment outcomes vary when the same antibiotics are used against different phenotypes of bacilli, and they are often ineffective. Most *M. abscessus* remain non-replicable in their low metabolic state within granulomas and form biofilms in the lung airways. It is an entirely different environment when compared to the testing of normal antibiotic activities. In this study, we tested the activity of the CC against *M. abscessus,* which survived in either biofilm or anaerobic environments as they are encountered in patients’ lung. The superiority of CC is that it is effective even under such disadvantageous biofilm conditions.

NTM can grow and survive extra- as well as intra-cellularly, for instance, inside the macrophages. In the context of NTM pulmonary infection, bacilli invade the mucosa and get phagocytized by macrophages. The infected macrophage cell lines may represent physiological conditions that mimic the real NTM disease scene while eradicating *M. abscessus*. As shown in Figure 5, CC inhibits the growth of intracellular *M. abscessus* in macrophage cell lines. We assume CC acts as protonophore uncouplers which disrupt the mycobacterial membrane and inhibit cell wall biosynthesis. As a result, CC regulates the growth of *M. abscessus*. Further studies for revealing the mode of actions are needed.

In conclusion, the results of this study on CC activity suggest that it may be used to develop new drug hypotheses against *M. abscessus* infections.

## 4. Materials and Methods

### 4.1. Bacterial Strains and Culture Conditions

The *M. abscessus* strain (ATCC 19997) was grown at 37 °C in Middlebrook 7H9 broth (BD, 27130) supplemented with 10% OADC (BD, 212240), 0.5% glycerol, and 0.05% Tween 80. We obtained the nine clinical strains from the Korean Collection of Type Collections. We created strains expressing green fluorescence protein and luciferase using the *M. abscessus* strain (ATCC 19997) for the intracellular activity assays. These strains were transformed by electroporation with pMV306hsp+LuxG13 (Addgene plasmid #26161) and pTEC15 (Addgene plasmid #30174).

*M. abscessus* (ATCC 19997) CLR-resistant mutants were selected on 7H11 (Sigma-Aldrich, M0428, (St. Louis, MO, USA)) agar plates containing 25, 50, and 100 times higher than the already known MIC value 0.1 ug/mL for CLR. 

To determine the antibiotic ability against *M. abscessus* (ATCC 19997) in anaerobic conditions, 5 × 10^6^ bacteria/mL were inoculated into 5 uL of 7H9 liquid medium (BD, 271310), supplemented with 10% OADC. Bacterial cultures were placed into anaerobic jars (BD BBLTM GasPak^TM^ Jar; Franklin Lakes, NJ, USA). We used BD BBL^TM^ GsaPak^TM^ anaerobic indicator (methylene blue used to monitor oxygen depletion). For drug susceptibility testing, bacteria were grown in cation-adjusted Mueller-Hinton broth (MH; Sigma-Aldrich, 90922, (St. Louis, MO, USA)).

### 4.2. Dose Response Curve Testing 

The MICs were determined according to the CLSI guidelines [10,11]. We used the REMA method CLSI recommended in MH with an inoculum of OD 0.001/mL in the exponential growth phase. We treated CC in actively growing *M. abscessus*, *M. abscessus*- pLUX G13, CLR-R mutant, and anaerobic cultured *M. abscessus*. The REMA assay is as follows. CC compound was diluted two-fold in a ten-point serial dilution in 96-well plates containing the bacilli in a total volume of 100 uL and then was incubated for three days at 37 °C before the addition of 40 uL of 0.025% resazurin. After overnight incubation, the fluorescence of resorufin (metabolite resazurin) was determined by using Synergy H1 Hybrid Multi-Mode Reader (Bio-Tek, Winooski, VT, USA). IC_50_ values were calculated from the raw fluorescence data using Prism 5.0 software (GraphPad Inc., La Jolla, CA, USA). The experiments were carried out with triplicates.

### 4.3. Biofilm Assays

The first biofilm assay system is the Calgary Biofilm Device employed by Bardouniotis et al. to evaluate the cidal activity or the minimal biofilm eradication concentration (MBEC) of biocides on *Mycobacterium phlei* [38,39]. We used MBEC Assay^®^ (Innovotech, Edmonton, AB, Canada) for measuring MBEC. The Biofilm formation was conducted as follows: first, 180 uL of 5 × 10^7^ bacterial inoculums per milliliter were distributed into a 96-well microtiter plate. Placing the 96-peg Lid onto a 96-well plate, it was then maintained for five days to form the biofilm on the peg. Next, 200 uL test drug solution prepared with medium was added to each well of the 96-well plate. Then, the 96-peg Lid was placed onto a 96-well plate and the plate was incubated for four days. Surviving bacteria in biofilms were removed from pegs by sonication in media followed by washing. Following incubation, the remaining adherent bacteria in biofilms with or without antibiotics were quantified by resazurin. 

### 4.4. Intracellular Killing Assay

THP-1 cells were treated with a final concentration of 50 nM Phorbol 12-Myristate 13-Acetate (PMA; Sigma Aldrich, St. Louis, MO, USA) for 48 h. Infection with *M. abscessus* harboring pTEC15 green fluorescent or luciferase was carried out at 37 °C in the presence of 5% CO_2_ for three hours at a multiplicity of resistance (MOI) of 2:1. After extensive washing with PBS, cells were incubated containing 50 ug/mL gentamycin for 30 min and washed again with PBS. Then 200 uL RPMI media containing DMSO (negative control) and 200 uL RPMI media containing the indicated concentration of CC were incubated for 3 to 4 days. The macrophages were stained with SYTO 60 (Invitrogen, Eugene, OR, USA) dye at a final concentration of 5 µM for 30 min at 37 °C, 5% CO_2_. Luminescence was measured on day four by using a Synergy H1 microplate reader (Bio-Tek, Winooski, VT, USA). Fluorescent images of live cells were captured by automated microscopy using a Lionheart™ FX automated microscopy (Bio-Tek, Winooski, VT, USA). The Gen5^TM^ 3.05 software object feature enables the identification of cells within the imaging field. 

### 4.5. Cell Viability Assay and Lactate Dehydrogenase (LDH) Cytotoxicity Assay

The cell viability and cytotoxicity of CC were evaluated in THP-1 (ATCC TIB-202) and HCT-8 cells (ATCC CCL-244) using the WST-8 Cell Viability Assay Kit (MediFab, Seoul, Korea) and CytoTox 96^®^ Non-Radioactive Cytotoxicity Assay (Promega, Madison, WI, USA), respectively. The differentiated THP-1 cells were placed into a 96-well plate (1.0 × 10^5^ cells/well) and HCT-8 was placed into a 96-well plate (2.0 × 10^4^ cells/well) and incubated at 37 °C for 24 h. We added CC to the cells at various concentrations. For the cell viability assay, 10 uL of reagent (10% media volume) was added to each well and incubated for four hours. The colors were measured at 450 nm. For the LDH cytotoxicity assay, 50 uL of LDH detection reagent was added to each well and incubated for 30 min in a dark room. The resulting color was measured at 490 nm using Synergy H1 microplate reader. We used cells treated with 1% Triton^TM^ X-100 as a positive control, while DMSO-treated cells were negative in both experiments.

### 4.6. Data Analysis

We processed data and constructed graphs with Prism version 7.0 (GraphPad) and Gen5^TM^ 3.05 software.

### 4.7. Ethics

All studies were approved by the Institutional Review Board of Masan National Tuberculosis Hospital (IRB-398837-2018-E34, approved on 14 January. 2019) and the Institutional Biosafety Committees (MTHIBC-19-01 and MTHIBC-21-11, approved on 26 Feburary. 2019 and 15 July 2021).

## Figures and Tables

**Figure 1 ijms-22-11029-f001:**
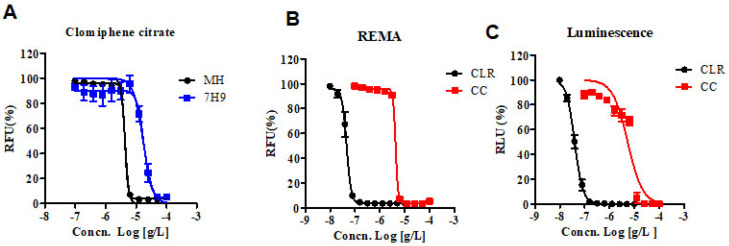
In vitro activity of CC. (**A**) The activity of CC against *M. abscessus* cultured in 7H9 broth with 10% OADC and cation-adjusted Mueller–Hinton medium (MH). (**B**) The activity of CC against *M. abscessus* in MH broth medium. Dose-response curves were plotted from the REMA. RFU, relative fluorescence units. CLR was used as a reference control. (**C**) The activity of CC against *M. abscessus*—Lux, incubated under the same condition as in panel B. RLU, relative luciferase units. The experiments were carried out with three biological replicates and expressed as the mean ± SEM for each concentration. This result was made from a representative experiment.

**Figure 2 ijms-22-11029-f002:**
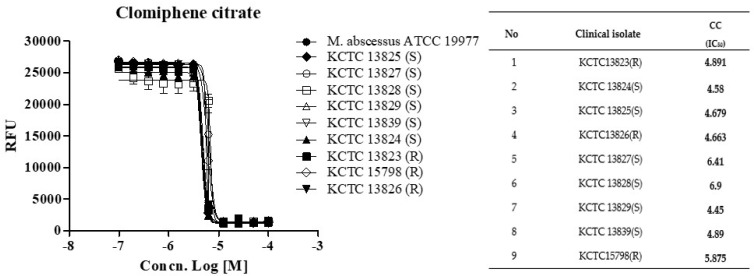
The activity of CC against *M. abscessus* clinical isolates. Dose-response curves (DRC) were plotted from the REMA data for isolates treated with CC. Data were expressed as the mean ± standard deviation (SD) of triplicates for each concentration. Analysis of DRC graph and IC_50_ using GraphPad Prism 5.0. RFU, relative fluorescence units. This result was made from a representative experiment.

**Figure 3 ijms-22-11029-f003:**
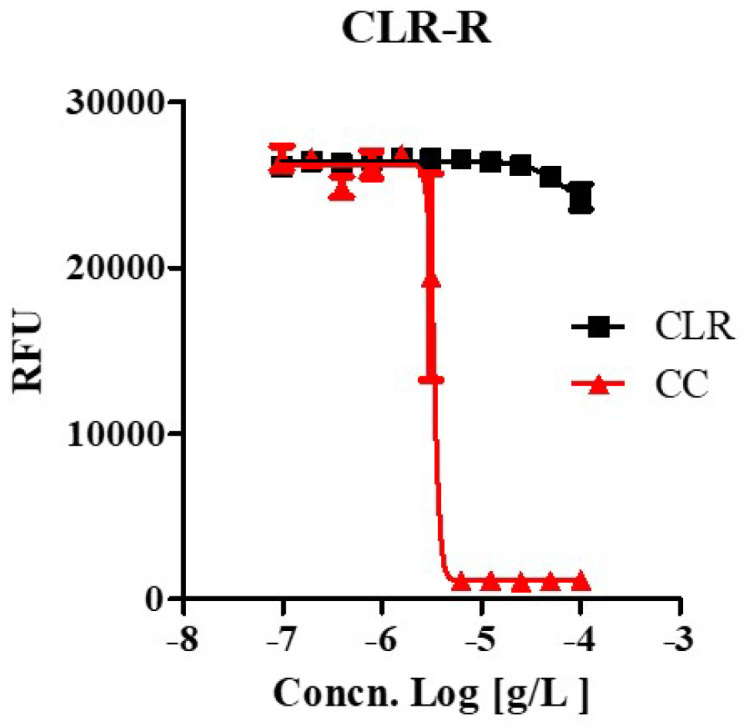
The activity of CC against clarithromycin-resistant *M. abscessus* mutants. Clarithromycin-resistant *M. abscessus* mutants (*M. abscessus* CLR-R) were tested for their ability to grow in Mueller-Hinton medium when treated with 100 ug/mL to 0.097 ug/mL of clarithromycin (CLR) and CC. Dose-response curves of *M. abscessus* CLR-R mutant (Left panel). The experiments were carried out with three biological replicates and expressed as the mean ± SEM for each concentration. This result was made from a representative experiment.

**Figure 4 ijms-22-11029-f004:**
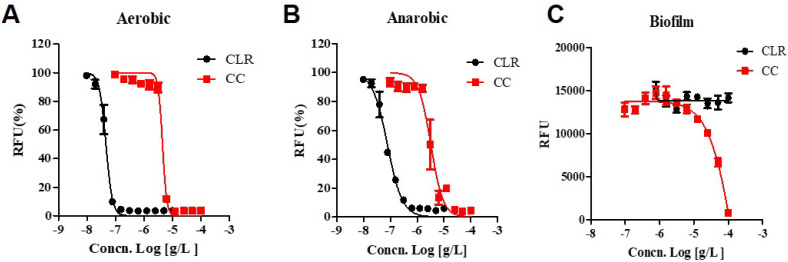
Dose-response curves of CC. Dose-response curves of *M. abscessus* under aerobic, anaerobic, and biofilm conditions using resazurin. CLR was used as a reference control. (**A**) aerobic condition, *M. abscessus* was cultured until the mid-log phase, taking approximately 24 h. (**B**) anaerobic condition, *M. abscessus* was undergone for seven days within an anaerobic generating container system (BD GasPak™EZ). (**C**) Biofilm, *M. abscessus* was examined with Calgary Biofilm Device. The experiments were carried out with three biological replicates and expressed as the mean ± SEM for each concentration. This result was made from a representative experiment.

**Figure 5 ijms-22-11029-f005:**
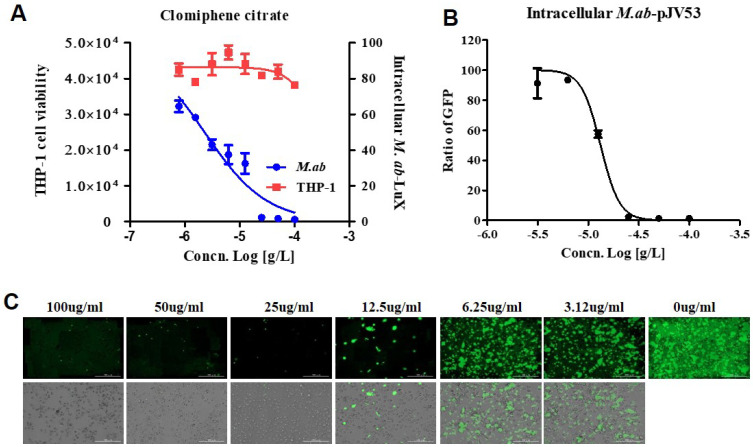
CC promotes the killing of intracellular *M. abscessus*. (**A**) Dual-readout assay of CC within macrophages. Infections in macrophages were carried out with multiplicities of infection (MOI) of two bacteria per cell for 3 h and washed to remove extracellular mycobacteria. Luminescence detected from luciferase-expressing *M. abscessus* alone within THP-1 macrophages. The infected THP-1 cells stained with SYTO 60 were counted using fluorescence detection, and total cells were counted using Multi Reader. (**B**) THP-1 activated with PMA were infected at MOI of 2 with GFP-expressed *M. abscessus* for 3 h, followed by treatment with CC-indicated concentrations in fresh medium. Ratio sum intensity values were used for all data reduction steps. (**C**) Images of GFP-expressing *M. abscessus* infected THP-1 cells on day 3 after treatment with indicated concentrations of CC. Automated microscopy using a Lionheart^TM^ automated live-cell imager. The Gen5^TM^ 3.05 software object feature enables the identification of cells within the imaging field. Data were expressed as the mean ± SEM of triplication for each concentration. (Scale bar = 100 um). This result was made from a representative experiment.

## Data Availability

Not applicable.

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
