# Peer review of "Clomiphene Citrate Shows Effective and Sustained Antimicrobial Activity against Mycobacterium abscessus"

_ijms, 2021, doi:10.3390/ijms222011029_

Round 1
Reviewer 1 Report
Comments and Suggestions for Authors are attached.

Author Response
Reviewer 1
- On the basis of what premises the authors selected clomiphene (an anti-estrogen compound that selectively inhibits the hypothalamic receptor binding estradiol, i.e. a compound that in fact stimulates ovulation) as a purpose of the current study (searching for a drug with antimicrobial activity)?
→→→ We were inspired by the paper by Lupien, Andréanne et al. and their finding that clomiphene was effective in treating tuberculosis.
- The authors have written that ‘CC appears to have few reported adverse side effects’. In my opinion, it should be noted what side effects may occur during or after treatment with clomiphene. I cannot agree that clomiphene can have only a few side effects. There is no information that clomiphene can cause visual disturbances, in some cases irreversible (this is, of course, a matter of the dose and duration of therapy), cerebrovascular incidents, psychotic reactions. A newer reference relating to the side effects of clomiphene would be better than a 2005 review.
→→→ Although there were mentions of side effects of CC in the papers published in 2013, they were about the other substances that improved CC while the papers published in 2005 describe the side effects of CC itself. This makes the paper from 2005 more relevant and therefore most recent.
- Latin names are spelled differently. Words such as ‘in vitro’, ‘M. abscessus’ etc. should be written consistently throughout the manuscript, in principle in italics.
→→→ I have corrected all the manuscript you pointed out.
- ‘Non-tuberculosis Mycobacteria’ or ‘Nontuberculous Mycobacteria’? - The full name of the abbreviation NTM is different in this manuscript. Authors should specify the full name of the NTM abbreviation.
→→→ The corresponding abbreviations were corrected.
- Throughout the manuscript, units (mg, ml, μg, etc.) are written in an unusual font or in a plain font (eg. Figure 5). The units should be written consistently throughout the manuscript.
→→→ I have corrected all the manuscript including those in the figures you pointed out. - Subsection 4.2: There is no reference to the CLSI guidelines. Authors should complete the missing reference.
→→→ I attached all the references you pointed out.
- The Figure 2 is illegible and should be corrected.
→→→ I've added a table to write down the IC50 values in the figure for easier understanding.

Reviewer 2 Report
In this work, Lee et al. have investigated the antimicrobial activity of Clomiphene citrate (CC) against Mycobacterium abscessus, the effects were compared to Clarithromycin treatment. The authors performed a series of experiments where the effects of CC on in vitro growth (culture medium) of a M. abscessus laboratory strain, clinical isolates, Clarithromycin-resistant M. abscessus as well as under aerobic, anaerobic and biofilm conditions. Finally, the effects of CC were tested on M. abscessus-infected cells by analyzing viability and bacterial luminescence in experiments with the THP1 macrophage cell line. The study is of interest as we lack effective treatments for infections with non-tuberculous mycobacteria.
However, the authors should address the following points:
- The manuscript is difficult to read and requires English language editing (Example, page 2: “There is a surge in research into experimental antibiotics with potential activity against M. abscessus in various mechanisms and new therapeutic treatments. However, there have not been sufficiently studied for the efficacy against M. abscessus.»). This is the case for large sections of the text, also the the result section is difficult to read.
- The following reference might be added after reference 8 on page 2: https://www.frontiersin.org/articles/10.3389/fmicb.2018.02541/full
- Abbreviations should be defined at their first appearance in the text, all abbreviations (e.g. REMA) should be defined
- The REMA assay should be described and termed as REMA in the methods section
- The terms MIC and IC50 should be used appropriately in the text and explained (IC50 was calculated from MIC experiments? Obtained MICs are not mentioned in text/figures?)
- How often and with how many replicates were figures produced? Values from one representative or several independent experiment shown? What show the error bars? Please state this information in the figure legend.
- Figures 1 and 4: Explain (main text and legend) why Clarithromycin was used.
- Figures 2 and 3 might be merged, the example image of the REMA could go to supplementary or should be shown in Figure 1 where this assay is first used.
- Figure 5: From the text it seems that the effects of Clarithromycin on THP1 viabiliy and M. abscessus replication in THP1 was assessed (“data not shown”). These are important controls and should be displayed in Figure 5A and 5B.
- It would have been interesting to see if combining CC and Clarithromycin shows possible synergistic effects.
- The effects of CC on non-tuberculous mycobacteria other than M. abscessus would have been of interest.
- The study lacks mechanistic data. Can the authors briefly discuss or speculate on possible mechanisms from what is known about CC from the literature (e.g. ref 8)?
- End of the discussion: The authors suggest CC as “potent new drug” for M. abscessus infection treatment. This cannot be concluded from in vitro growth-inhibition studies and one cell line infection experiment. The authors should be modest in their conclusions. The same goes actually for the title, where the antimicrobial activity against M. abscessus should be highlighted rather than implicating CC as promising new treatment.
Author Response
In this work, Lee et al. have investigated the antimicrobial activity of Clomiphene citrate (CC) against Mycobacterium abscessus, the effects were compared to Clarithromycin treatment. The authors performed a series of experiments where the effects of CC on in vitro growth (culture medium) of a M. abscessus laboratory strain, clinical isolates, Clarithromycin-resistant M. abscessus as well as under aerobic, anaerobic and biofilm conditions. Finally, the effects of CC were tested on M. abscessus-infected cells by analyzing viability and bacterial luminescence in experiments with the THP1 macrophage cell line. The study is of interest as we lack effective treatments for infections with non-tuberculous mycobacteria.
However, the authors should address the following points:
- The manuscript is difficult to read and requires English language editing (Example, page 2: “There is a surge in research into experimental antibiotics with potential activity against M. abscessus in various mechanisms and new therapeutic treatments. However, there have not been sufficiently studied for the efficacy against M. abscessus.»). This is the case for large sections of the text, also the the result section is difficult to read.
→→→ The overall structure of the text mentioned was modified by asking a native speaker.
- The following reference might be added after reference 8 on page 2: https://www.frontiersin.org/articles/10.3389/fmicb.2018.02541/full
→→→ I have added all the references you pointed out.
- Abbreviations should be defined at their first appearance in the text, all abbreviations (e.g. REMA) should be defined
→→→ I have corrected all the abbreviations you pointed out.
- The REMA assay should be described and termed as REMA in the methods section
→→→ The REMA method has been described in the methods section.
- The terms MIC and IC50 should be used appropriately in the text and explained (IC50 was calculated from MIC experiments?
→→→ The IC50 value was calculated using a prism program and indicated in the legend.
- How often and with how many replicates were figures produced? Values from one representative or several independent experiments shown? What show the error bars? Please state this information in the figure legend.
→→→ Basically, all experiments were repeated 3 times. The standard error mean value is displayed in the legend of all the figures.
- Figures 1 and 4: Explain (main text and legend) why Clarithromycin was used.
→→→ Clarithromycin was used as a control because it is the most representative drug for treating M. abscessus.
- Figures 2 and 3 might be merged, the example image of the REMA could go to supplementary or should be shown in Figure 1 where this assay is first used.
→→→ Figure 2 is the experimental result for clinical isolates while figure 3 is the experimental result for the clarithromycin resistant mutant, so it was explained separately. REMA experimental images have been moved to supplement 1.
- Figure 5: From the text it seems that the effects of Clarithromycin on THP1 viability and M. abscessus replication in THP1 was assessed (“data not shown”). These are important controls and should be displayed in Figure 5A and 5B.
→→→ The results of experiments with clarithromycin have already been published several times in references. We focused on explaining how effective CC is in THP1.
As described in the text, the IC50 value of clarithromycin is 0.1 ㎍/㎖, so it is difficult to express it in one graph, so it is omitted.
- It would have been interesting to see if combining CC and Clarithromycin shows possible synergistic effects.
→→→ Unfortunately, in this study, we did not do a combining CC with Clarithromycin. It’s a very interesting point. We will be sure to include it in future experiments.
- The effects of CC on non-tuberculous mycobacteria other than M. abscessus would have been of interest.
→→→ As unpublished data, the result of our experiments with NTMs other than M.abscessus, M.avium and M.intraelluare did not show good effects.
- The study lacks mechanistic data. Can the authors briefly discuss or speculate on possible mechanisms from what is known about CC from the literature (e.g. ref 8)?
→→→ In this study, the possibility of CC was confirmed. After reporting the results of this experiment, we are planning a study on the mechanism of CC. We will continue to do research.
- End of the discussion: The authors suggest CC as “potent new drug” for M. abscessus infection treatment. This cannot be concluded from in vitro growth-inhibition studies and one cell line infection experiment. The authors should be modest in their conclusions. The same goes actually for the title, where the antimicrobial activity against M. abscessus should be highlighted rather than implicating CC as promising new treatment.
→→→ The title of the paper was revised according to the reviewer's suggestion, and the word potent was deleted from discussion and replaced with another word. It is hoped that this report will lead to the development of a new NTM drug based on the results of in vitro and intracellular experiments.

Round 2
Reviewer 2 Report
The manuscript improved and the authors addressed most of my comments.
However, there are still some minor points that I would like to see addressed:
- The figure legends in the revised manuscript state that experiments were "carried out with three replicates and expressed as the mean ± SEM for each concentration" (suggesting one experiment). In their response letter the authors state that "Basically, all experiments were repeated 3 times. The standard error mean value is displayed in the legend of all the figures" (this suggests 3 independent experiments). This is confusing. The authors should clearly state somewhere if the figures were made from one (representative) experiment and if and how often the experiments were repeated.
- On my comment on Figure 4 (use of Clarithromycin in the figure), authors replied in their letter that this was used as control. This was clear to me, but for the reader this information should be added to the result text and Figure 1 and 4 legend.
- I appreciate that the REMA example image has been moved to the supplementary data file. However, the description/title of the Figure S2 (p12) should provide information that Figure S1 shows an example image of the REMA.
- Figure 5 in which authors do not show data that "CLR successfully rescued the cells from the infection challenge and prevented bacterial replication (data not shown)".
In their reply, the authors state that the "IC50 value of clarithromycin is 0.1 ㎍/㎖, so it is difficult to express it in one graph". From the shown graphs in Fig. 5 I do not see why this is difficult. In addition, the authors reply that "results of experiments with clarithromycin have already been published several times in references". In this case, the authors should at least list these references where they refer to "data not shown" on page 7.
Author Response
Reviewer 2
The manuscript improved and the authors addressed most of my comments.
However, there are still some minor points that I would like to see addressed:
- The figure legends in the revised manuscript state that experiments were "carried out with three replicates and expressed as the mean ± SEM for each concentration" (suggesting one experiment). In their response letter the authors state that "Basically, all experiments were repeated 3 times. The standard error mean value is displayed in the legend of all the figures" (this suggests 3 independent experiments). This is confusing. The authors should clearly state somewhere if the figures were made from one (representative) experiment and if and how often the experiments were repeated.
This figure was made from representative experiment.
→→→ I have corrected all figures you pointed out.
- On my comment on Figure 4 (use of Clarithromycin in the figure), authors replied in their letter that this was used as control. This was clear to me, but for the reader this information should be added to the result text and Figure 1 and 4 legend.
→→→ I have added the information in the legend of figure 1 and 4.
- I appreciate that the REMA example image has been moved to the supplementary data file. However, the description/title of the Figure S2 (p12) should provide information that Figure S1 shows an example image of the REMA.
→→→ I have corrected description/title of the figure S1 you pointed out.
- Figure 5 in which authors do not show data that "CLR successfully rescued the cells from the infection challenge and prevented bacterial replication (data not shown)".
In their reply, the authors state that the "IC50 value of clarithromycin is 0.1 ㎍/㎖, so it is difficult to express it in one graph". From the shown graphs in Fig. 5 I do not see why this is difficult. In addition, the authors reply that "results of experiments with clarithromycin have already been published several times in references". In this case, the authors should at least list these references where they refer to "data not shown" on page 7.
→→→ I have added the representative reference at ref. 19 as you pointed out and described them in the text.

Round 3
Reviewer 2 Report
I thank the authors for addressing my additional comments